# Zhang–Zhang Polynomials of Multiple Zigzag Chains Revisited: A Connection with the John–Sachs Theorem

**DOI:** 10.3390/molecules26092524

**Published:** 2021-04-26

**Authors:** Henryk A. Witek

**Affiliations:** 1Department of Applied Chemistry, Institute of Molecular Science, National Yang Ming Chiao Tung University, 1001 Ta-Hsueh Road, Hsinchu 300092, Taiwan; hwitek@mail.nctu.edu.tw; 2Center for Emergent Functional Matter Science, National Yang Ming Chiao Tung University, 1001 Ta-Hsueh Road, Hsinchu 300092, Taiwan

**Keywords:** enumeration of Clar covers, ZZ polynomials, Clar covering polynomials, benzenoids, multiple zigzag chains, John–Sachs theorem

## Abstract

Multiple zigzag chains $Z\left(m,n\right)$ of length *n* and width *m* constitute an important class of regular graphene flakes of rectangular shape. The physical and chemical properties of these basic pericondensed benzenoids can be related to their various topological invariants, conveniently encoded as the coefficients of a combinatorial polynomial, usually referred to as the ZZ polynomial of multiple zigzag chains $Z\left(m,n\right)$. The current study reports a novel method for determination of these ZZ polynomials based on a hypothesized extension to John–Sachs theorem, used previously to enumerate Kekulé structures of various benzenoid hydrocarbons. We show that the ZZ polynomial of the $Z\left(m,n\right)$ multiple zigzag chain can be conveniently expressed as a determinant of a Toeplitz (or almost Toeplitz) matrix of size $\left\lceil \frac{m}{2}\right\rceil \times \left\lceil \frac{m}{2}\right\rceil$ consisting of simple hypergeometric polynomials. The presented analysis can be extended to generalized multiple zigzag chains $Z_{k}\left(m,n\right)$, i.e., derivatives of $Z\left(m,n\right)$ with a single attached polyacene chain of length *k*. All presented formulas are accompanied by formal proofs. The developed theoretical machinery is applied for predicting aromaticity distribution patterns in large and infinite multiple zigzag chains $Z\left(m,n\right)$ and for computing the distribution of spin densities in biradical states of finite multiple zigzag chains $Z\left(m,n\right)$.

## 1. Introduction

The construction, enumeration, and characterization of Kekulé structures and Clar covers of benzenoid hydrocarbons constitute one of the most important topics of graph-theoretical characterization of chemical molecules, a scientific discipline dwelling in the intersection of theoretical chemistry and discrete mathematics [1,2,3,4,5]. For a chemist, a Kekulé structure is a resonance structure in which each carbon atoms participates in exactly one double bond [6]. Similarly, a Clar cover is a resonance structure in which each carbon atoms participates in exactly one double bond or in exactly one aromatic sextet with some other five carbon atoms [7]. These concepts are explained graphically in Figure 1 using pyrene (i.e., the multiple zigzag chain structure $Z\left(2,2\right)$) as an example, together with an intuitive definition of the ZZ polynomial of $Z\left(2,2\right)$. For a mathematician, a Kekulé structure *K* is a spanning subgraph of the molecular graph *G* corresponding to a given benzenoid B, such that every component of *K* is isomorphic to a complete graph on two vertices, $K_{2}$. Similarly, a Clar cover *C* is a spanning subgraph of *G*, such that every component of *C* is isomorphic to $K_{2}$ or isomorphic to a cyclic graph on six vertices, $C_{6}$ [8]. The most important questions pertinent to graph-theoretical characterization of a given benzenoid B are as follows. (i) What is the number of Kekulé structures, $K\equiv K\left(B\right)$, that can be constructed for B? (ii) What is the number of Clar covers, $C\equiv C\left(B\right)$, that can be constructed for B? (iii) What is the maximal number, $Cl\equiv Cl\left(B\right)$, of aromatic sextets $C_{6}$ that can be accommodated in B? (iv) How many distinct Clar covers exist, $c_{Cl}\equiv c_{Cl}\left(B\right)$, with exactly Cl aromatic sextets? (v) Assuming that *k* is a non-negative integer, how many distinct Clar covers exist, $c_{k}\equiv c_{k}\left(B\right)$, with exactly *k* aromatic sextets? Note that the answer to the last question, i.e., the sequence of numbers $\left[c_{0},c_{1},c_{2},\ldots \right]$, is the most general here, being capable of also providing a solution to the preceding questions. Indeed, we have $K=c_{0}$, $C=\sum_{k}c_{k}$, $Cl=\max\left\{k|c_{k}\neq 0\right\}$, and $c_{Cl}=c_{\max\left\{k|c_{k}\neq 0\right\}}$. Note that for a finite benzenoid B with *N* atoms, the maximal number of aromatic sextets that can be accommodated in B is naturally bounded from above by N/6, and thus for all k>N/6, we have $c_{k}=0$. The most convenient way of representing the subsequence $\left[c_{0},c_{1},c_{2},\ldots ,c_{Cl}\right]$ is given in the form of its generating function
(1)$$\mathrm{ZZ}\left(B,x\right)=\sum_{k=0}^{Cl}c_{k}\;x^{k},\;$$
which is most often referred to as the Clar covering polynomial or, from the names of its inventors, as the Zhang–Zhang polynomial or the ZZ polynomial of B [9,10,11,12,13,14,15]. Substantial research effort has been invested in the determination of $\mathrm{ZZ}\left(B,x\right)$ for elementary families of benzenoids [8,16,17,18,19,20,21,22,23,24,25,26,27,28,29,30,31,32,33,34,35,36,37,38,39,40,41]. The rapid development of Clar theory stimulated by these discoveries in recent years has led to many new interesting applications and connections to other branches of chemistry, graph theory, and combinatorics [8,17,18,19,21,28,34,42,43,44,45,46,47,48,49,50,51,52,53,54,55,56,57,58,59,60,61].

In one of our recent contributions to the field of characterization of Clar covers of basic pericondensed benzenoids, we have derived [33] a closed-form generating function for the ZZ polynomials of multiple zigzag chains $Z\left(m,n\right)$ and generalized multiple zigzag chains $Z_{k}\left(m,n\right)$ [2,62]. For a graphical definition of both families of benzenoids, see Figure 2. For $Z\left(m,n\right)$, the derived formula (Equation (Equation 20) of the work in [33]) had an aesthetically pleasing and compact structure of a finite, regular continued fraction
(2)$$\sum_{m=0}^{\infty }\mathrm{ZZ}\left(Z\left(m,n\right),x\right)t^{m}=\left[0;-t,{\left(-1\right)}^{2}z\;t,{\left(-1\right)}^{3}z\;t,\ldots ,{\left(-1\right)}^{n}z\;t,1+{\left(-1\right)}^{n+1}z\;t\right]$$
where, as usual in the theory of ZZ polynomials, z=1+x, and where the continued fraction notation $\left[a_{0};a_{1},a_{2},a_{3},\ldots ,a_{n}\right]$ should be interpreted as in the following example
(3)$$\left[a_{0};a_{1},a_{2},a_{3},a_{4}\right]=a_{0}+\frac{1}{a_{1}+\frac{1}{a_{2}+\frac{1}{a_{3}+\frac{1}{a_{4}}}}}$$

The formula in Equation (Equation 2) is quite regular, but probably its high symmetry is most elegantly reflected in a slightly different continued fraction representation:(4)∑m=0∞ZZZm,n,xtm=−1zt+−1zt+−1⋱+⋱zt+−1zt+−1zt−−1n
Note that the length of the continued fraction in Equation (Equation 4) is n+1, i.e., the number of times the product zt appears in it is *n*. Similar expressions, in terms of products of generating functions of the form (Equation 4), were obtained for the generalized multiple zigzag chains $Z_{k}\left(m,n\right)$ shown in Figure 2.

Despite the internal beauty of the $\mathrm{ZZ}\left(Z\left(m,n\right),x\right)$ and $\mathrm{ZZ}\left(Z_{k}\left(m,n\right),x\right)$ generating functions, it turned out that extracting the original ZZ polynomials from the generating functions in Equations (Equation 2) and (Equation 4) is a rather formidable task. A binomial expansion of the continued fraction of type (Equation 4), starting from the most shallow level, together with an appropriate multiple sum rotation produces a rather lengthy and cumbersome expression for the ZZ polynomial of $Z\left(m,n\right)$ given by
(5)ZZZm,n,x=∑l1=0m∑l2=0l1⋯∑ln=0ln−1∏j=2n+1−1ljcj−lj−1lj−1−ljzl1
with the coefficient $c_{j}$ expressed by
(6)$$c_{j}={\left(-1\right)}^{j}\left(1+m+2\sum_{i=1}^{j-2}{\left(-1\right)}^{i}l_{i}\right)$$

Similar expansions for $\mathrm{ZZ}\left(Z_{k}\left(m,n\right),x\right)$ were not rigorously attempted, in conviction that their mathematical structure is much too complicated and of little practical importance. Equations (Equation 5) and (Equation 6) provide formally a closed-form formula for $\mathrm{ZZ}\left(Z\left(m,n\right),x\right)$, but at the same time, they do not bring full intellectual satisfaction of possessing such a formula, as Equation (Equation 5) provides little, if any, conceptual insight into the problem of enumeration of Clar covers of $Z\left(m,n\right)$. We concluded our previous work [33] by expressing a hope that the evaluation of multiple sums in Equation (Equation 5) could lead to a more compact and transparent expression, but to date no such a result has been reported.

In the current work, we report the discovery and demonstrate formal validity of new ZZ polynomial formulas for the multiple zigzag chains $Z\left(m,n\right)$ and the generalized multiple zigzag chains $Z_{k}\left(m,n\right)$. The new formulas possess both a transparent algebraic form and deep internal structure, which reveal a direct connection to the structural parameters *m*, *n*, and *k*, and offer a possibility for fast and robust determination of the ZZ polynomials in a form of a determinant of $\left(n+2\right)$-diagonal Toeplitz (or almost Toeplitz) matrices of size $\left\lceil \frac{m}{2}\right\rceil \times \left\lceil \frac{m}{2}\right\rceil$ or $\left\lceil \frac{m+1}{2}\right\rceil \times \left\lceil \frac{m+1}{2}\right\rceil$. The new formulas are numerically consistent with the previously obtained Equations (Equation 5) and (Equation 6). In a sense, the reported formulas are a direct conceptual extension of the formulas in Equations (14.34)–(14.36) and Equations (14.38)–(14.39) of the seminal work of Cyvin and Gutman [2], which reported the determinantal John–Sachs expressions for the number of Kekulé structures for $Z\left(m,n\right)$ and $Z_{k}\left(m,n\right)$. The actual form of the reported Toeplitz matrices reported here originates from an anticipated extension of the John–Sachs theorem to the world of Clar covers. This hypothesized extension encouraged us to recently conduct an intensive search of examples of elementary benzenoids B, for which it is possible to construct a generalization of the John–Sachs matrix, whose determinant would be equal to the ZZ polynomial of B and which, upon the evaluation x=0, would reduce to the original John–Sachs matrix of B. Among other examples given in [63], we have been able to achieve this task for multiple zigzag chains with n=1, 2, and 3. Here, we have been able to achieve this goal for general values of the structural parameters *m* and *n* for $Z\left(m,n\right)$, as well as for an arbitrary-sized structures $Z_{k}\left(m,n\right)$. In addition, the somewhat *ad hoc* derivations of these formulas are accompanied here with a formal mathematical proof of their validity using the usual recurrence relations obeyed simultaneously by the ZZ polynomials of $Z\left(m,n\right)$ and $Z_{k}\left(m,n\right)$ and the determinants of the discovered Toeplitz matrices.

In the last section of the current manuscript, we present an application of the developed theoretical framework for computing the aromaticity distribution patterns and the biradical spin population patterns in finite and infinite multiple zigzag chains $Z\left(m,n\right)$. These structures, usually referred by physicists as graphene nanoribbons, received considerable attention from the physics community as promising candidates for future nanoelectronics and spintronics [64,65,66,67,68].

## 2. Preliminaries

To explain our reasoning leading to the discovery of determinantal formulas for the ZZ polynomials of the multiple zigzag chains $Z\left(m,n\right)$ and generalized multiple zigzag chains $Z_{k}\left(m,n\right)$, we need to start with some formal preliminaries. Assume, like always, that a benzenoid B is a planar graph embedded in a hexagonal lattice. We define a Kekulé structure *K* of B as a spanning subgraph of B whose components are $K_{2}$ (i.e., complete graphs on 2 vertices). Similarly, we define a Clar cover *C* of B as a spanning subgraph of B whose components are either $K_{2}$ or $C_{6}$ (i.e., cycle graphs of length 6) [7]. The number of the $C_{6}$ components in a given Clar cover *C* will be referred to as the order of this Clar cover. Denoting by $c_{k}$ the number of distinct Clar covers of order *k*, the generating function for the sequence $\left[c_{0},c_{1},c_{2},\ldots ,c_{Cl}\right]$ is usually referred to as the Clar covering polynomial (or Zhang–Zhang polynomial or ZZ polynomial) of B and is formally defined as
(7)$$\mathrm{ZZ}\left(B,x\right)=\sum_{k=0}^{Cl}c_{k}\;x^{k},$$
where the maximal number Cl of the $C_{6}$ components that can be accommodated within B is referred to as the Clar number of B [9,10,11,12,13,14,15]. In practice, the ZZ polynomial of an arbitrary benzenoid B can be robustly computed using recursive decomposition algorithms [4,20,26] or can be conveniently determined using interface theory of benzenoids [31,39,69,70,71,72]. A useful theoretical tool for finding ZZ polynomials for an arbitrary benzenoid is ZZDecomposer [25,26]. With this program, one can conveniently define a benzenoid using a mouse drawing pad and subsequently use the underlying graph representation of the benzenoid to find its ZZ polynomial, manipulate its Clar covers, and determine its structural similarity to other related benzenoids. ZZDecomposer has been successfully applied to find close-form formulas of ZZ polynomials for numerous families of basic elementary benzenoids [20,22,23,24,25,27,29,30,32,33,34,36,37,63,73,74].

Assume further that the benzenoids $Z\left(m,n\right)$ and $Z_{k}\left(m,n\right)$ are drawn in a plane in the way shown in Figure 3 with some of their edges oriented vertically. We say that a vertex $p_{i}$ is a *peak*, if all its neighbors are located below $p_{i}$. Similarly, we say that a vertex $v_{j}$ is a *valley*, if all its neighbors are located above $v_{j}$. It is a well-known fact that if the structure B is Kekuléan, and both families of $B=Z\left(m,n\right)$ and $B=Z_{k}\left(m,n\right)$ possess this quality for any admissible values of the structural parameters *m*, *n*, and *k*, then the number of peaks in B is equal to the number of valleys in B; let us denote this number as *l*. It is clear from Figure 3 that $l=\frac{m}{2}$ for $Z\left(m,n\right)$ with even *m*, $l=\frac{m+1}{2}$ for $Z\left(m,n\right)$ and $Z_{k}\left(m,n\right)$ with odd *m*, and $l=\frac{m+2}{2}$ for $Z_{k}\left(m,n\right)$ with even *m*. The next two important concepts are the *wetting region* of a peak $p_{i}$, which is defined as a subgraph of B spanned by all the vertices of B that are accessible from $p_{i}$ by going exclusively downward, and the *funnel region* of a valley $v_{j}$, which is defined as a subgraph of B spanned by all the vertices of B that are accessible from $v_{j}$ by going exclusively upward. Finally, the $p_{i}\to v_{j}$
*path region* is defined as the intersection of the wetting region of $p_{i}$ and the funnel region of $v_{j}$. All these concepts [75] are illustrated schematically in Figure 4.

Let us finally denote by Kpi→vj the number of Kekulé structures for the pi→vj path region. Then, for an arbitrary Kekuléan benzenoid B with *l* peaks p1,…,pl and *l* valleys v1,…,vl, the John–Sachs matrix ℙB is defined as a square, l×l matrix with elements Pij=Kpi→vj. The celebrated John–Sachs theorem (see, for example, Theorem 1 in [76] or Equation (5) in [77]) states that the number of Kekulé structures for B is equal to the determinant of ℙB,
(8)KB=ℙB.

In our recent work [63], we have asked whether the concept of the John–Sachs matrix ℙB can be generalized to encompass the theory of Clar covers. The natural extension meant to make the matrix ℙB
*x*-dependent, i.e., replacing Pij=Kpi→vj by Pij=ZZpi→vj,x, does not work, as the determinant of the resulting matrix does not evaluate to the ZZ polynomial of B. Notwithstanding, we have discovered that for numerous families of basic benzenoids, it is actually possible to find a generalized John–Sachs matrix ℙZZB that possesses all the qualities expected for such a generalization:The determinant of the John–Sachs matrix ℙB is equal to the number of Kekulé structures as stipulated by Equation (Equation 8). Similarly, we request that the determinant of the generalized John–Sachs matrix ℙZZB is equal to the ZZ polynomial of B
(9)ℙZZB=ZZB,x.As the Kekulé structures of B are simply the Clar covers of B of order 0, the number KB of Kekulé structures can be obtained from the ZZ polynomial of B by evaluating it at x=0,
(10)ZZB,xx=0=KB,
which effectively corresponds to removing from the set of Clar covers those which contain at least one component C6. It is then only natural to request that in the same limiting process the John–Sachs path matrix ℙB is obtained from the generalized John–Sachs path matrix ℙZZB upon the evaluation at x=0
(11)ℙZZBx=0=ℙB.

Numerous examples of generalized John–Sachs path matrices ℙZZB have been given in our recent work [63]. Here, we augment this collection with two further examples corresponding to multiple zigzag chains Zm,n and generalized multiple zigzag chains Zkm,n. Note that most of the previous results were given without a proof, while here for both Zm,n and Zkm,n we are able to furnish formal proof of all of the presented results. Interestingly, the off-diagonal elements of the generalized John–Sachs path matrices ℙZZB have a rather unexpected form, for which no plausible explanation nor interpretation has been yet discovered. We hope that a formal extension of the John–Sachs theory to the world of Clar covers will explain this conundrum. We also hope that the results given in the current work will stimulate the community to work on such the generalization of John–Sachs theorem to the world of Clar covers. We finally truly hope that the designed here strategy will be sufficient to discover one of the last missing gems in the theory of ZZ polynomials of basic benzenoids, the ZZ polynomial formula for an arbitrary hexagonal flake Ok,m,n, similarly like the original John–Sachs theory was used to prove analogous formula for KOk,m,n.

## 3. Discovery of the Determinantal Formulas

In this section, we briefly discuss the process that led us to the discovery of the determinantal formulas of the ZZ polynomials of multiple zigzag chains Zm,n and generalized multiple zigzag chains Zkm,n. Our motivation for presenting this somewhat obsolete reasoning is purely pedagogical, as it is hoped that similar heuristic reasoning processes can lead to a discovery of other missing formulas in the theory of ZZ polynomials. We limit our discussion to the case of Zm,n with an even value of *m*. In our previous work [63], we have discovered that the (multiple) zigzag chains Zm,1, Zm,2, and Zm,3 are characterized by the following generalized John–Sachs path matrices:ℙZZZm,1=w0w10⋯0z⋱⋱⋱⋮0⋱⋱⋱0⋮⋱⋱⋱w10⋯0zw0⏟matrixm2×m2withw0=1+2zw1=z
ℙZZZm,2=w0w1w200z⋱⋱⋱⋮0⋱⋱⋱w2⋮⋱⋱⋱w10⋯0zw0⏟matrixm2×m2withw0=1+4z+z2w1=3z1+23zw2=z2ℙZZZm,3=w0w1w2w30z⋱⋱⋱w30⋱⋱⋱w2⋮⋱⋱⋱w10⋯0zw0⏟matrixm2×m2withw0=1+6z+3z2w1=6z1+43z+16z2w2=5z21+25zw3=z3
where, as before, z=1+x. A number of regularities can be immediately noticed:The matrices ℙZZZm,n are m2×m2 Toeplitz matrices with n+2 diagonals: 1 subdiagonal z, 1 diagonal w0, and *n* consecutive superdiagonals w1,…,wn.The value on the subdiagonal, z=1+x, does not depend on *n*.The value of the diagonal element w0 is equal to ZZM2,n,x, given explicitly by
(12)ZZM2,n,x=∑k=022knkzk=2F1−2,−n1;z.This value is consistent with the natural extension of the John–Sachs matrix ℙB diagonal elements, i.e., with replacing Pii=Kpi→vi by Pii=ZZpi→vi,x.The value wl on the *l*-th superdiagonal is a product of a multiplicative factor zl, a numerical factor cnl, and a polynomial pnlz of degree 2, 1, or 0.

Further analysis requires slightly larger amount of data. Similar techniques to those used in our previous work [63] allow us to find the values of wl for ℙZZZm,4 and ℙZZZm,5; thus, we have
(13)w0=1+8z+6z2w1=10z1+63z+12z2w2=15z21+45z+115z2w3=7z31+27zw4=z4forℙZZZm,4w0=1+10z+10z2w1=15z1+83z+1z2w2=35z21+65z+15z2w3=28z21+47z+128z2w4=9z31+29zw5=z5forℙZZZm,5

Equipped with this knowledge it is relatively easy to identify the last two missing pieces of the puzzles: the numerical factor $c_{nl}$ and the polynomial $p_{nl}\left(z\right)$.

The polynomials $p_{nl}\left(z\right)$ always start from 1 and contain coefficients, which factorize into small primes. This suggests that they are hypergeometric polynomials, i.e., hypergeometric functions ${}_{p}F_{q}\left[\begin{array}{c} a_{1},\ldots ,a_{p}\\ b_{1},\ldots ,b_{q} \end{array};z\right]$ with at least one of the upper indices $a_{j}$ being a negative integer. As the formula for the diagonal elements in Equation (Equation 12) contains a hypergeometric function ${}_{2}F_{1}$, it is natural to seek $p_{nl}\left(z\right)$ also in this form. The lower parameter $b_{1}$ is suggested by the denominator in the linear term of the polynomials $p_{nl}\left(z\right)$, and it is equal to 3 for $w_{1}$, 5 for $w_{2}$, 7 for $w_{3}$, etc. Therefore, in a general case, we can expect for $w_{l}$ a value of $b_{1}=1+2l$. Another observation is that the coefficient in the linear term of the polynomial, equal to $\frac{a_{1}a_{2}}{b_{1}}$, is always positive, which, taking into account that at least one of the upper indices should be negative, shows that actually both $a_{1}$ and $a_{2}$ are negative. One of these numbers, say $a_{1}$, is immediately recognizable as −2, because of the constant degree of the polynomials $p_{nl}\left(z\right)$ for l≤n−2. The other index, $a_{2}$, is a function $a_{2}=a_{2}\left(n,l\right)$, which for l=n should be 0 (because of the degree 0 of the polynomial $p_{nl}\left(z\right)$ for l=n) and for l=n−1 should be 1 (because of the degree 0 of the polynomial $p_{nl}\left(z\right)$ for l=n). All these facts suggest that $a_{2}=l-n$. A straightforward verification with Maple [78] shows that indeed the hypergeometric function ${}_{2}F_{1}\left[\begin{array}{c} -2,-n+l\\ 1+2l \end{array};z\right]$ reproduces all the reported polynomials $p_{nl}\left(z\right)$ in the matrix elements $w_{l}$ given above. Note that this polynomial reproduces somewhat fortuitously also the diagonal entry $w_{0}$.The last remaining task is the identification of the two-dimensional sequence of numbers $c_{nl}$, which numerical values for small *n* and *l* are given by the following triangle:
(14)1111211311311615114110151711511535281911n/l012345This task can be readily performed by typing, for example, the last rows of this triangle in the The On-Line Encyclopedia of Integer Sequences [79], which recognizes it as the sequence A085478 generated by
(15)cnl=n+l2l≡n+ln−l

These two identifications conclude our heuristic deduction and allow to express the ZZ polynomial of multiple zigzag chains Zm,n with even *m* as a determinant of the generalized John–Sachs path matrix ℙZZZm,n
(16)ZZZm,n,x=ℙZZZm,n=w0⋯wn00z⋱wm⋱⋮0⋱⋱wmwn⋮⋱⋱⋱⋮0⋯0zw0⏟determinantm2×m2,
where the matrix elements $w_{l}$ are given explicitly by
(17)wl=1+xln+ln−l2F1−2,−n+l1+2l;1+x.

The hypergeometric function in Equation (Equation 17) can be expanded as a power series in $\left(1+x\right)$, which gives a binomial-like definition convenient for evaluation:(18)wl=∑j=022jn+l2l+j1+xl+j.

While the definition of $w_{l}$ given by Equation (Equation 18) is probably more transparent and practical, we want to stress that the hypergeometric form in Equation (Equation 17) has been essential in the process of the identification of $w_{l}$, furnishing a convenient unified framework for the search process.

It is further possible to make the form of the matrix ℙZZZm,n slightly more uniform by noticing that the presence of the binomial coefficient n+ln−l in Equation (Equation 17) (or the binomial n+l2l+j in Equation (Equation 18)) imposes that
(19)wl=0forl>n,
which means we could omit the triangle of zeros in Equation (Equation 16) and simply fill all the superdiagonals with the entries wl. We also note in passing that Equation (Equation 18) naturally extends also to l=−1, giving
(20)w−1=1+x,
which allows to rewrite the entry z=1+x on the first subdiagonal of Equation (Equation 16) simply as w−1. This gives the most transparent form of the ZZ polynomial of a multiple zigzag chain Zm,n with even *m* as
(21)ZZZm,n,x=ℙZZZm,n=w0w1w2⋯wm−22w−1⋱⋱⋱⋮0⋱⋱⋱w2⋮⋱⋱⋱w10⋯0w−1w0⏟determinantm2×m2,
where the matrix elements wl are computed using Equation (Equation 18).

The correctness of Equation (Equation 16) has been extensively tested by comparing it with the ZZ polynomials of multiple zigzag chains $Z\left(m,n\right)$ computed by brute-force recursive calculations using ZZDecomposer for various values of the structural parameters *m* and *n*. A formal proof of correctness of Equation (Equation 16) is presented in Section 4.

So far, we have discussed the determinantal formula for the ZZ polynomial of multiple zigzag chains Zm,n with even *m*. Now, we explain how analogous formulas for Zkm,n and for Zm,n with odd values of *m* have been discovered. It is probably simplest to start with Zkm,n with even *m*. Figure 3 suggests that for even *m*, the generalized John–Sachs matrix ℙZZZkm,n corresponds to a matrix of size m2+1×m2+1. As, for even values of *m*, the first m2 peaks and m2 valleys of Zkm,n coincide with those of Zm,n, it is natural to expect that the first, diagonal m2×m2 block of ℙZZZkm,n is identical to ℙZZZm,n. The remaining diagonal entry corresponds to the pm+22→vm+22 path and according to the hypothesized generalization of the John–Sachs theorem it can be expressed as ZZM1,k,x=1+k(1+x). It is also natural to assume the value on the first subdiagonal still is z=1+x. Discovering the values on the superdiagonals requires intensive numerical experimentation similar in character to the efforts needed to discover wl. Fortunately, this process is relatively straightforward, as independent variations of two structural parameters—*k* and *m*—supply a sufficient amount of numerical data to complete the identification in an almost linear fashion. It turns out that the value vlk located on the lth superdiagonal in the last column of ℙZZZkm,n with even *m* can be expressed by the following hypergeometric formula:(22)vls=1+xls+lk−l2F1−1,−s+l1+2l;1+x
or its binomial-like equivalent
(23)vls=∑j=011js+l2l+j1+xl+j,
where s=k. Both formulas bear close resemblance to Equations (Equation 17) and (Equation 18). It is obvious from the presence of the binomial coefficient s+lk−l in Equation (Equation 22) that
(24)$$v_{l}^{s}=0\;\;\;\;\;\;\mathrm{for}\;\;\;\;\;\;l>s,$$
which means that the last column contains only k+1 non-zero entries, including the diagonal element. Somewhat fortuitously, the diagonal element, $v_{0}^{k}=1+kz$, can be also expressed by Equations (Equation 22) and (Equation 23) with l=0 and s=k.

No effort is needed to discover the form of the generalized John–Sachs matrix ℙZZZm,n for Zm,n with odd *m*; it is immediately identified as ℙZZZnm−1,n, because of the structural identity Zm,n=Znm−1,n.

The last remaining structure, for which we need to find the generalized John–Sachs matrix, is Zkm,n, with an odd value of *m*. This is a considerable harder task. Here, ℙZZZkm,n corresponds to a matrix of size m+12×m+12. Again, for odd values of *m*, the first m−12 peaks and m−12 valleys of Zkm,n coincide with those of Zm−1,n, so it is natural to expect that the first, diagonal m−12×m−12 block of ℙZZZkm,n is identical to ℙZZZm−1,n. The remaining diagonal element u0k corresponds to the pm+22→vm+22 path area, which has a shape of an elementary benzenoid, a ribbon Rbk,n−k,1,1, with the ZZ polynomial given by 1+n+k1+x+k22n−k−11+x2. (for details, see Equation (10) in [35]). Extensive numerical experimentation allows to establish that the value ulk located on the lth superdiagonal in the last column of ℙZZZkm,n with odd *m* can be expressed by
(25)ult=vln+1+x∑j=1tvln−j
where t=k, with vln and vln−j given by Equations (Equation 22) or (Equation 23). Direct evaluation of Equation (Equation 25) gives the following hypergeometric representation of this term:(26)ult=1+xln+l2l2F1−2,−n+l1+2l;1+x−1+xl+1n−t+l2l+12F1−1,−n+t+l+12+2l;1+x
or its binomial equivalent
(27)ult=∑j=022jn+l2l+j1+xl+j−∑j=011jn−t+l2l+1+j1+xl+1+j.

Explicit determinantal formulas for ZZ polynomials of multiple zigzag chains $Z\left(m,n\right)$ and generalized multiple zigzag chains $Z_{k}\left(m,n\right)$ discovered in the previous paragraphs have been presented in Figure 2 together with binomial representation of all the matrix elements required for their determination and graphical representation of the analyzed structures. Figure 2 can be thought as the graphical abstract of the current work summarizing the most important discoveries reported here. For those who prefer hypergeometric representation of the elements of the relevant path matrices, the following compact formulas should suffice: (28)wl=1+xln+l2l2F1−2,−n+l1+2l;1+x,(29)vls=1+xls+l2l2F1−1,−s+l1+2l;1+x,(30)ult=wl−1+xl+1n−t+l2l+12F1−1,−n+t+l+12+2l;1+x.

## 4. Formal Proof of Determinantal Formulas for $\mathrm{ZZ}\left(Z\left(m,n\right),x\right)$ and $\mathrm{ZZ}\left(Z_{k}\left(m,n\right),x\right)$

We start the proof by demonstrating that the formulas presented in Figure 2 produce the ZZ polynomials of $Z\left(m,n\right)$ and $Z_{k}\left(m,n\right)$ for low values of the index *m*. We argue in the next paragraph that ZZ polynomials of multiple zigzag chains $Z\left(m,n\right)$ with m≥3 and ZZ polynomials of generalized multiple zigzag chains $Z_{k}\left(m,n\right)$ with m≥1 can be computed in a recursive fashion from analogous ZZ polynomials of $Z\left(m,n\right)$ and $Z_{k}\left(m,n\right)$ with lower values of the index *m*. Such a recursive algorithm works neither for $Z\left(m,n\right)$ with m=1 or m=2 nor for $Z_{k}\left(m,n\right)$ with m=0, so we treat these cases separately here. Simple geometric considerations show that $Z\left(2,n\right)=M\left(2,n\right)$, $Z\left(1,n\right)=M\left(1,n\right)$, and $Z_{k}\left(0,n\right)=M\left(1,k\right)$. As the ZZ polynomial of a parallelogram $M\left(m,n\right)$ is well known (ZZMm,n,x=∑j=0mmjnj1+xj, see for example Equation (4) of [24]), we obtain the following relations:(31)ZZZ1,n,x=ZZM1,n,x=∑j=011jnj1+xj=v0n=ℙZZZ1,n,
(32)ZZZ2,n,x=ZZM2,n,x=∑j=022jnj1+xj=w0=ℙZZZ2,n,
(33)ZZZk0,n,x=ZZM1,k,x=∑j=011jkj1+xj=v0k=ℙZZZk0,n,
which confirm that the ZZ polynomials of $Z\left(1,n\right)$, $Z\left(2,n\right)$, and $Z_{k}\left(0,n\right)$ can be computed from the corresponding generalized John–Sachs matrices presented in Figure 2.

For $Z\left(m,n\right)$ with m≥3 and for $Z_{k}\left(m,n\right)$ with m≥1, the formal proof is a consequence of the general recursive relations derived by Zhang and Zhang for ZZ polynomials. For the multiple zigzag chains $Z\left(m,n\right)$ and generalized multiple zigzag chains $Z_{k}\left(m,n\right)$, the relevant recursive relations are derived in Figure 5 by choosing a particular edge in $Z\left(m,n\right)$ and $Z_{k}\left(m,n\right)$ and assigning it to $K_{2}$, $C_{6}$, or none. The three possibilities are labeled in Figure 5 as D, R, and S, respectively, where the designated letter abbreviates the corresponding chemical terms: a double bond, an aromatic ring, and a single bond. The resulting recurrence relations interconnecting the ZZ polynomials of various structures $Z\left(m,n\right)$ and $Z_{k}\left(m,n\right)$
(34)$$\begin{array}{ccc} \mathrm{ZZ}\left(Z\left(m,n\right),x\right) & = & \mathrm{ZZ}\left(Z_{n-1}\left(m-1,n\right),x\right)+\left(1+x\right)\cdot \mathrm{ZZ}\left(Z\left(m-2,n\right),x\right) \end{array}$$
(35)$$\begin{array}{ccc} \mathrm{ZZ}\left(Z_{k}\left(m,n\right),x\right) & = & \mathrm{ZZ}\left(Z_{k-1}\left(m,n\right),x\right)+\left(1+x\right)\cdot \mathrm{ZZ}\left(Z_{n-k}\left(m-1,n\right),x\right) \end{array}$$
should be consistently satisfied for all the permissible values of the structural parameters *m*, *n*, and *k*. Noting that $Z\left(m,n\right)\equiv Z_{0}\left(m,n\right)\equiv Z_{n}\left(m-1,n\right)$, we see that Equation (Equation 34) is a specialized version of Equation (Equation 35) with k=n and *m* replaced by m−1. It is therefore sufficient to consider only Equation (Equation 35), which is valid for the following values of the structural parameters: 1≤m and 1≤k≤n together with the initial conditions given by Equations (Equation 31)–( Equation 33) and corresponding to the edge cases $Z\left(1,n\right)$, $Z\left(2,n\right)$, and $Z_{k}\left(0,n\right)$. Equipped with this input, we are able to demonstrate below that the deduced earlier determinantal formulas for $\mathrm{ZZ}\left(Z\left(m,n\right),x\right)$ and $\mathrm{ZZ}\left(Z_{k}\left(m,n\right),x\right)$ (see Figure 2) satisfy the recurrence relation in Equation (Equation 35).

### 4.1. $Z\left(m,n\right)$ with Even m

Let us compute both sides of Equation (Equation 34) assuming the correctness of the determinantal formulas given in Figure 2. LHS of Equation (Equation 34) evaluates to ℙZZZm,n given by Equation (Equation 21). RHS of Equation (Equation 34) evaluates to

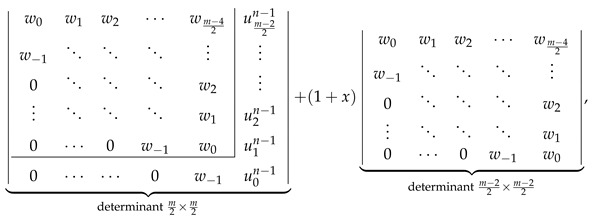
(36)
where the value of $u_{l}^{n-1}$ defined in Equation (Equation 27) evaluates to the following explicit expression
(37)$$\begin{array}{ccc} u_{l}^{n-1} & = & \left\{\begin{array}{cc} w_{0}-\left(1+x\right) & \;\mathrm{for}\;l=0,\\ w_{l} & \;\mathrm{for}\;l>0.\; \end{array}\right. \end{array}$$

Substituting the values of ul given by Equation (Equation 37) to the first determinant in Equation (Equation 36) and splitting it with respect to the last column into a sum of two determinants, (see, for example, in [80]) allows us to rewrite Equation (Equation 36) as a sum of ℙZZZm,n and

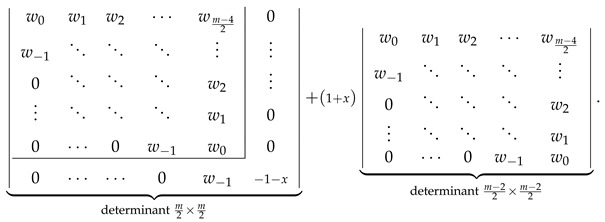
(38)

Laplace expansion of the first determinant in Equation (Equation 38) with respect to the last column shows that the both addends in Equation (Equation 38) cancel each other, retaining only ℙZZZm,n as the evaluation of RHS of Equation (Equation 34). This is identical to the LHS of Equation (Equation 34), proving the consistency of the determinantal formulas reported here with the recurrence relation in Equation (Equation 34) for even *m*.

### 4.2. $Z\left(m,n\right)$ with Odd m

The analogous consistency demonstration for the $\mathrm{ZZ}\left(Z\left(m,n\right),x\right)$ determinantal formulas with odd *m* is slightly more complicated. We start with LHS of Equation (Equation 34) evaluated to the determinantal form with odd *m*,

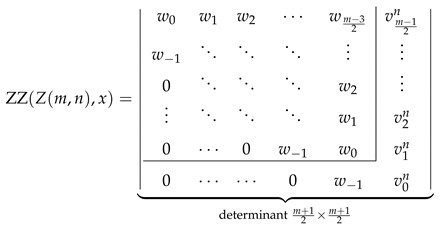
(39)
and perform the following consecutive operations on the following determinant:Use $v_{l}^{s}$ given by Equation (Equation 23) to rewrite the determinant in Equation (Equation 39) by expressing its last column as a+b, where $a={\left[v_{\frac{m-1}{2}}^{n}-v_{\frac{m-1}{2}}^{n-1},\ldots ,v_{1}^{n}-v_{1}^{n-1},v_{0}^{n}-v_{0}^{n-1}\right]}^{⊺}$ and $b={\left[v_{\frac{m-1}{2}}^{n-1},\ldots ,v_{1}^{n-1},v_{0}^{n-1}\right]}^{⊺}$.Decompose the determinant into a sum of two determinants differing only by the last column [80]

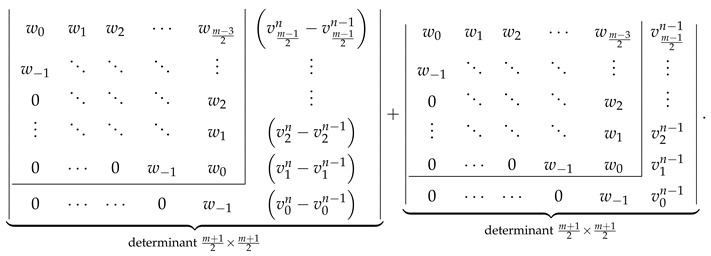
(40)Identify $\left(⋆\right)$ the second determinant in Equation (Equation 40) as $\mathrm{ZZ}\left(Z_{n-1}\left(m-1,n\right),x\right)$ with the help of Figure 2.Rewrite the first determinant in Equation (Equation 40) by subtracting its last column from its last-but-one column obtaining

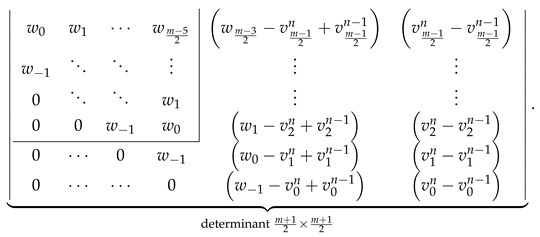
(41)Direct calculations using the definitions in Equations (Equation 18) and (Equation 23) allow us to establish that
$$\begin{array}{ccc} v_{0}^{n}-v_{0}^{n-1} & = & 1+x,\\ w_{-1}-v_{0}^{n}+v_{0}^{n-1} & = & 0,\\ w_{l}-v_{l+1}^{n}+v_{l+1}^{n-1} & = & v_{l}^{n}\;\;\;\mathrm{for}\;\;\;l>0. \end{array}$$Using these relations simplifies the determinant in Equation (Equation 41) to the following form:

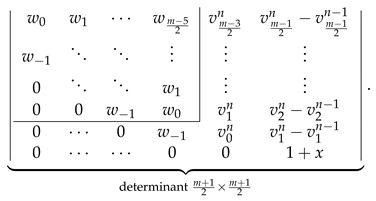
(42)Laplace expansion of the determinant in Equation (Equation 42) with respect to the last row gives

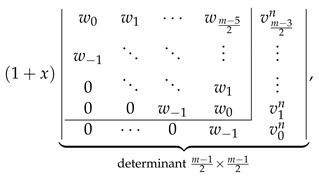
(43)
allowing us to identify $\left(⋆\;⋆\right)$ the expression in Equation (Equation 43) as $\left(1+x\right)\mathrm{ZZ}\left(Z\left(m-2,n\right),x\right)$.

The sum of both performed identifications, $\left(⋆\right)$ and $\left(⋆\;⋆\right)$, is equal to the RHS of Equation (Equation 34) evaluated to the determinantal form with odd *m*; this equality confirms the consistency of the determinantal formulas presented in Figure 2 with Equation (Equation 34) for odd *m*.

### 4.3. $Z_{k}\left(m,n\right)$ with Even m

The consistency check process in this case is quite similar to that discussed in the previous section, so we minimize the discussion, giving only its most important intermediate steps. We start with the LHS of Equation (Equation 35) evaluated for even *m* to the determinantal form with the help of Figure 2 obtaining

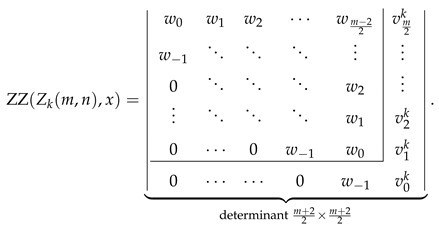
(44)

We perform the following consecutive operations on this determinant:We rewrite the last column of the determinant in Equation (Equation 44) as a+b, with $a_{l}=v_{l}^{k}-v_{l}^{k-1}$ and $b_{l}=v_{l}^{k-1}$.We decompose the determinant in Equation (Equation 44) as a sum of two determinants, the first one having a in the last column and the second one having b in the last column.We identify $\left(⋆\right)$ the second determinant with the help of Figure 2 as $\mathrm{ZZ}\left(Z_{k-1}\left(m,n\right),x\right)$.In the first determinant, we subtract the last column from the last-but-one column obtaining

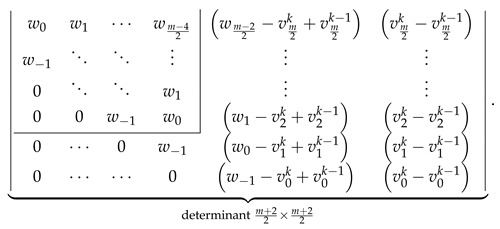
(45)Direct calculations using the definitions in Equations (Equation 18), (Equation 23) and (Equation 27) allow us to establish that
$$\begin{array}{ccc} v_{0}^{k}-v_{0}^{k-1} & = & 1+x,\\ w_{-1}-v_{0}^{k}+v_{0}^{k-1} & = & 0,\\ w_{l}-v_{l+1}^{k}+v_{l+1}^{k-1} & = & u_{l}^{n-k}\;\;\;\mathrm{for}\;\;\;l>0; \end{array}$$
using these relations simplifies the determinant in Equation (Equation 45) to the following form:

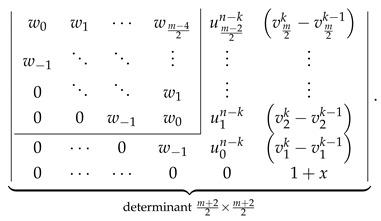
(46)Laplace expansion of the determinant in Equation (Equation 46) with respect to the last row gives

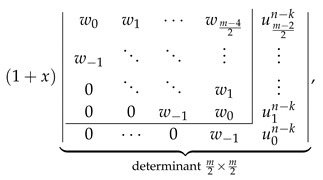
(47)
allowing to identify $\left(⋆\;⋆\right)$ this expression as $\left(1+x\right)\mathrm{ZZ}\left(Z_{n-k}\left(m-1,n\right),x\right)$.

The sum of both performed identifications, $\left(⋆\right)$ and $\left(⋆\;⋆\right)$, representing LHS of Equation (Equation 35) by construction, is identical to RHS of Equation (Equation 35) evaluated to the determinantal form for even *m* using the formulas presented in Figure 2.

### 4.4. $Z_{k}\left(m,n\right)$ with Odd m

The consistency check process in this case relies on the identity $u_{l}^{k}=u_{l}^{k-1}+\left(1+x\right)v_{l}^{n-k}$ easily derivable directly from the definition of $u_{l}^{k}$ given by Equation (Equation 25). We start with LHS of Equation (Equation 35) evaluated to the determinantal form for odd *m* with the help of Figure 2, obtaining

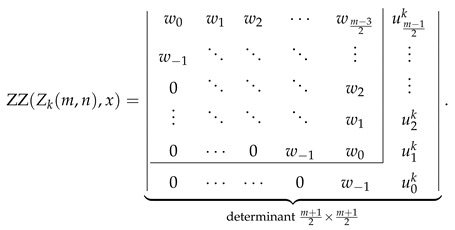
(48)

Substituting the aforementioned identity for for every $u_{l}^{k}$ in the last column of the determinant in Equation (Equation 48) and representing it as a sum of two determinants (decomposition with respect to the last column) gives

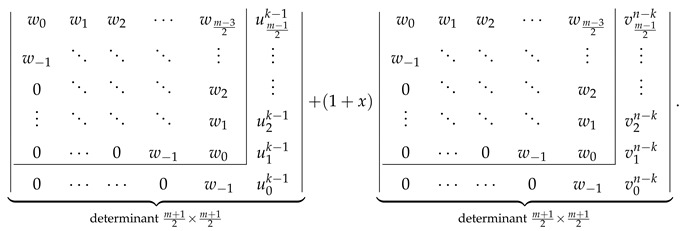
(49)

The first determinant in Equation (Equation 49) is identified using Figure 2 as $\mathrm{ZZ}\left(Z_{k-1}\left(m,n\right),x\right)$, and the second determinant as $\mathrm{ZZ}\left(Z_{n-k}\left(m-1,n\right),x\right)$. Altogether, the expression in Equation (Equation 49) reproduces the RHS of Equation (Equation 35) evaluated to the determinantal form for odd *m*, concluding the consistency check for all possible values (odd and even) of *m* for both types of structures, $Z\left(m,n\right)$ and $Z_{k}\left(m,n\right)$, and consequently proving correctness of the determinantal formulas of multiple zigzag chains and generalized multiple zigzag chains given in Figure 2.

## 5. Chemical Applications

In the current section, we use the derived equations to determine various chemical properties of multiple zigzag chains $Z\left(m,n\right)$. In particular, we focus on the aromaticity pattern distribution within a multiple zigzag chain $Z\left(m,n\right)$ and on the distribution of unpaired spins in biradical excited states of multiple zigzag chains $Z\left(m,n\right)$. First, we define the ZZ aromaticity indicator and the ZZ spin populations and compare them with analogous quantities obtained from rigorous quantum chemical calculations using selected systems. Subsequently, the ZZ indices are computed for families of isostructural multiple zigzag chains and their variation with growing size of the graphene flake is analyzed. The developed ZZ methodology allows one to treat substantially larger flakes than the conventional quantum chemical calculations and in some cases permits computing the asymptotics in transition to infinite graphene nanoribbons.

### 5.1. Local ZZ Aromaticity Indicators for Multiple Zigzag Chains $Z\left(m,n\right)$

The ZZ aromaticity index for some hexagon $h_{i}$ of a multiple zigzag chain $Z\left(m,n\right)$ is defined as a ratio of the number of Clar covers of $Z\left(m,n\right)$ in which the hexagon $h_{i}$ has aromatic character to the total number of Clar covers of $Z\left(m,n\right)$. The denominator in this ratio is easily computed as $\mathrm{ZZ}\left(Z\left(m,n\right),1\right)$, while the hexagon-dependent numerators must be computed independently for each hexagon of $Z\left(m,n\right)$. The maximal hexagon aromaticity computed in this way for an arbitrary graphene flake cannot exceed the theoretical value characterizing the infinite graphene sheet, i.e., 33%; the minimal value has a natural lower bound of 0%. The absolute aromaticity scale obtained in this way is very convenient, as it not only allows for finding most and least aromatic hexagons within a given graphene flake, but it also allows for aromaticity comparisons between flakes with various shapes.

The ZZ aromaticity indices computed in this way are shown schematically in Figure 6 for two multiple zigzag chains: $Z\left(5,3\right)$ and $Z\left(4,6\right)$. The maximal ZZ hexagon aromaticity in $Z\left(5,3\right)$ is approximately 67% of the value expected for the infinite graphene sheet, while for $Z\left(4,6\right)$, the maximal value is smaller (45%). A few general observations can be made. (i) Various quantum chemical aromaticity indicators show rather large discrepancies in the predicted aromaticity patterns. In particular, the Bird aromaticity indicator differs substantially from all the other indicators. (ii) The ZZ aromaticity pattern for $Z\left(5,3\right)$ agrees quite well with analogous patterns predicted by HOMA, NICS, PDI, and FLU, i.e., with the majority of selected quantum chemical aromaticity indicators. (iii) The ZZ aromaticity pattern for $Z\left(4,6\right)$ is similar to the patterns predicted by majority of other indicators (NICS, PDI, FLU, SCI, and PLR). (iv) The hexagons with maximal aromaticity in $Z\left(5,3\right)$ and $Z\left(4,6\right)$ are correctly predicted by the ZZ approach. (v) The hexagons with minimal aromaticity (i.e., with maximal antiaromaticity) in $Z\left(5,3\right)$ and $Z\left(4,6\right)$ are again correctly predicted by the ZZ approach. In particular, the ZZ approach is the only one to indicate very clear antiaromatic character of the hexagons characterized by positive chemical shift (blue dots in Figure 6) in the NICS approach. The two benchmark comparisons presented in Figure 6 suggest that the ZZ polynomial approach to graphene flakes aromaticity can be very useful, allowing to treat systems much larger than those accessible with quantum chemical calculations, as the computational cost of computing ZZ polynomials is only fractional in comparison to quantum chemical calculations.

The ZZ approach is now used to study the convergence of the aromaticity patterns for the family of multiple zigzag chains $Z\left(5,n\right)$ with the growing value of *n*. In Figure 7, we present the computed ZZ aromaticity indicators for all the hexagons of $Z\left(5,n\right)$ with n=3,6,9,12,15,21, and 45. A few observations are in place. (i) The aromaticity pattern in $Z\left(5,n\right)$ seems to converge with *n* to a uniformly non-aromatic infinite graphene strip. (ii) The location of the most aromatic and the most anti-aromatic hexagons is not altered by the elongation of the flake. (iii) In each of the five polyacene chains in $Z\left(5,n\right)$, the aromaticity decreases monotonically from the convex end to the concave end. By studying the recursive decomposition of $Z\left(5,n\right)$, it is possible to derive an analytical formula for the aromaticity of the most aromatic hexagon in each polyacene row. Denoting by $A_{k}\left(n\right)$ the aromaticity of the most aromatic hexagon in the row *k* of $Z\left(5,n\right)$, the results of the analysis can be given as follows:(50)$$\begin{array}{ccc} A_{1}\left(n\right)=A_{5}\left(n\right) & = & \frac{\mathrm{ZZ}\left(Z_{n-1}\left(3,n\right),1\right)}{\mathrm{ZZ}\left(Z\left(5,n\right),1\right)}\;\\ \; & = & \frac{5(10n^{4}+20n^{3}+8n^{2}-2n-3)}{\left(1+2n\right)\left(32n^{4}+64n^{3}+68n^{2}+36n+15\right)} \end{array}$$
(51)$$\begin{array}{ccc} A_{2}\left(n\right)=A_{4}\left(n\right) & = & \frac{\mathrm{ZZ}\left(Z\left(1,n\right),1\right)\cdot \mathrm{ZZ}\left(Z_{n-1}\left(2,n\right),1\right)}{\mathrm{ZZ}\left(Z\left(5,n\right),1\right)}\;\\ \; & = & \frac{5\left(2n+3\right){\left(2n-1\right)}^{2}}{\left(32n^{4}+64n^{3}+68n^{2}+36n+15\right)} \end{array}$$
(52)$$\begin{array}{ccc} \;\;\;\;\;\;\;\;\;\;\;\;\;\;\;\;\;\;\;\;\;\;\;\;\;\;\;\;\;\;\;\;\;\;A_{3}\left(n\right) & = & \frac{{\left(\mathrm{ZZ}\left(Z_{n-1}\left(1,n\right),1\right)\right)}^{2}}{\mathrm{ZZ}\left(Z\left(5,n\right),1\right)}\;\\ \; & = & \frac{15{\left(2n^{2}+2n-1\right)}^{2}}{\left(1+2n\right)\left(32n^{4}+64n^{3}+68n^{2}+36n+15\right)} \end{array}$$

Each of these formulas converges like c/n to 0 when n→∞ with the coefficient *c* equal to 25/32 for $A_{1}\left(n\right)$, 40/32 for $A_{2}\left(n\right)$, and 30/32 for $A_{3}\left(n\right)$, showing that an infinitely long graphene strip of width 5 is completely non-aromatic, i.e., that all the bonds have either single or double character. At the same time, the formulas show that in each finite nanoribbon $Z\left(5,n\right)$ the most aromatic hexagon is located at the convex end of the second polyacene row. Note that obtaining these conclusions would not be possible directly from quantum chemical calculations without the developed here ZZ polynomial methodology.

### 5.2. Spin Densities in Biradical Multiple Zigzag Chains $Z\left(5,n\right)$

Another application of the developed here methodology is the computation of spin densities in biradical multiple zigzag chains $Z\left(5,n\right)$. The probability of finding an unpaired electron on the $p_{z}$ orbital of a selected carbon atom is computed from the associated ZZ polynomial in a similar manner as described above for aromaticities. Namely, we consider all possible distributions of two unpaired electrons in a biradical $Z\left(5,n\right)$ structure by localizing them at the $p_{z}$ orbitals of two selected carbon atoms—C${}_{i}$ and C${}_{j}$, and for each such distribution, we compute the number of Clar covers corresponding to it by evaluating the associated ZZ polynomial at x=1 (or equivalently, at z=2). This number, divided by the total number of Clar covers in the non-radical $Z\left(5,n\right)$ structure, gives the probability of finding the two unpaired electrons at positions C${}_{i}$ and C${}_{j}$ simultaneously. Summing such contributions over all the possible distributions of two unpaired electrons gives the total atomic probabilities of finding an unpaired electron over each carbon atom. The probabilities obtained in this way are then normalized to 1 and the resulting ZZ spin density probability distributions are plotted as red (spin up) and blue (spin down) circles centered at carbon atoms; for more details, see in [92]. The results computed in this manner are plotted in Figure 8 for two families of multiple zigzag chains: $Z\left(5,n\right)$ with n=4,5,6,7,8,9,10,12,15 and 21 (upper panel) and $Z\left(m,10\right)$ with m=4,5,6 and 7 (lower panel). All the presented spin density distributions look alike and consist of a single ↑ spin wave at one zigzag edge of $Z\left(m,n\right)$ and a single ↓ spin wave at the other zigzag edge of $Z\left(m,n\right)$. The spin densities at the carbon atoms in the interior of $Z\left(m,n\right)$ are negligible, except for the smallest studied here structures. The spin waves at each zigzag edge have no nodal points with the maximum of the distribution located approximately at the middle of the edge. By extrapolating this picture to infinite graphene strips, one can expect that the zigzag edges of the strip will have a constant spin density for each carbon atom located at each zigzag edge, with one side corresponding to spin up, and the other side, to spin down. The resulting spin density pattern is consistent with the predicted previously edge-state magnetism of infinite graphene nanoribbons with zigzag edges, for which it was suggested that the antiferromagnetic state may at some conditions be lower in energy than the nonmagnetic state, leading to spontaneous magnetization of such nanoribbons [64,93,94,95,96,97]. Similar considerations for finite-length nanoribbons did not confirm this effect, suggesting that the ferromagnetic correlation of unpaired electrons at the edge of finite nanoribbons is rather an excited state property [98,99]. Nevertheless, it is remarkable that simple topological considerations performed with Clar covers and ZZ polynomials can lead to similar conclusions as those drawn from sophisticated *ab initio* quantum mechanical calculations.

## 6. Conclusions

Inspired by the recently reported collection of generalized John–Sachs matrices ℙZZB for various classes of elementary aromatic hydrocarbons B [63], which allowed us to express the ZZ polynomial of B simply as ZZB,x=ℙZZB, we have extended this approach in the current work to two important families of benzenoids: multiple zigzag chains Zm,n and generalized multiple zigzag chains Zkm,n, defined graphically in Figure 2. The extension allowed us to discover compact and simple determinantal formulas for ZZZm,n,x and ZZZkm,n,x, offering a possibility of considerable simplifications in the process of computation of ZZ polynomials for these two types of structures. The previously known ZZZm,n,x formula, reported originally as Equation (58) in [33] and reproduced here as Equation (Equation 5), has the form of multiple sums of multiple products of complicated binomials and is too complex to shed light on the structure and classification of Clar covers of Zm,n. In contrast, the determinantal formulas reported here hide all the algebraic complexity in the form of a determinant of a very simple object: a highly-structured Toeplitz or almost-Toeplitz matrix. In particular, we have discovered that for multiple zigzag chains Zm,n with even values of the structural parameter *m*, their ZZ polynomials have particularly transparent form given by
(53)ZZZm,n,x=w0w1w2⋯wm−22w−1⋱⋱⋱⋮0⋱⋱⋱w2⋮⋱⋱⋱w10⋯0w−1w0⏟determinantm2×m2,
where wl is a simple polynomial in 1+x given explicitly by
(54)wl=∑j=022jn+l2l+j1+xl+j.
Note that $w_{l}\equiv 0$ for l>n, so maximally only the lowest n+2 diagonals do not vanish identically. An analogous formula for the ZZ polynomial of multiple zigzag chains $Z\left(m,n\right)$ with odd values of the structural parameter *m* is only slightly more complicated

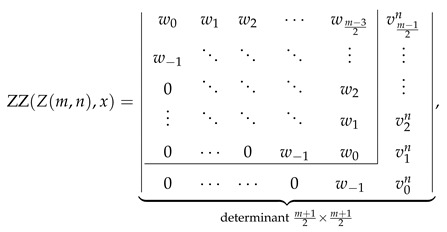
(55)
where $v_{l}^{s}$ is again a simple polynomial in $\left(1+x\right)$ given explicitly by
(56)vls=∑j=011js+l2l+j1+xl+j.

The corresponding formulas for $\mathrm{ZZ}\left(Z_{k}\left(m,n\right),x\right)$ closely follow the same pattern; we have

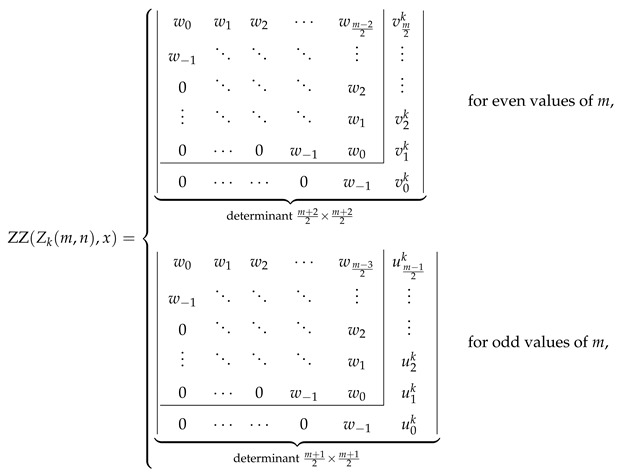
(57)
where $u_{l}^{t}$ is again a low-order polynomial in $\left(1+x\right)$ explicitly given by
(58)$$u_{l}^{t}=v_{l}^{n}+\left(1+x\right)\sum_{j=1}^{t}v_{l}^{n-j}.$$

The reported determinantal ZZ polynomial formulas for Zm,n and Zkm,n also implicitly define the corresponding generalized John–Sachs matrices ℙZZZm,n and ℙZZZkm,n for these two families of elementary benzenoids, which upon the evaluation x=0 reduce to the regular John–Sachs matrices ℙZm,n and ℙZkm,n given previously by Equations (14.33)–(14.41) in [2] (see also [62]). We want to stress that in contrast to our earlier work on generalized John–Sachs matrices [63] where most of the discovered facts were only conjectures; here, all the presented results are furnished with appropriate formal proofs. We hope that the presented here results will stimulate the graph-theoretical community to discover further examples of generalized John–Sachs matrices ℙZZ also for other benzenoids, which will pave the road to conception and formulation of the generalization of John–Sachs theorem [75,76,77,100,101,102,103,104,105,106] to the world of Clar covers. We also hope that the presented here techniques will suggest an appropriate line of attack on the most difficult unsolved problem in the theory of ZZ polynomials: discovering the closed-form formula for the ZZ polynomial of hexagonal flake Ok,m,n with arbitrary set of parameters [37,41,107].

The developed mathematical machinery of ZZ polynomials for $Z\left(m,n\right)$ and $Z_{k}\left(m,n\right)$ has been applied to two practical chemical problems: determination of aromaticity patterns in finite and infinite graphene nanoribbons and in the determination of spin populations in biradical states of graphene nanoribbons. These results show good agreement with previously reported quantum chemical data, reproducing similar aromaticity patterns for $Z\left(5,3\right)$ and $Z\left(4,6\right)$ as other, well-established aromaticity indicators and similar antiferromagnetic spin patterns as those obtained from solid state calculations. In contrast to the quantum chemical calculations, the developed here methodology allows for studying the transition from the molecular to crystal regime and establishing that the infinite zigzag ribbons are completely antiaromatic. Note that similar conclusions could not be reached from a standard quantum chemical view point.

## Figures and Tables

**Figure 1 molecules-26-02524-f001:**
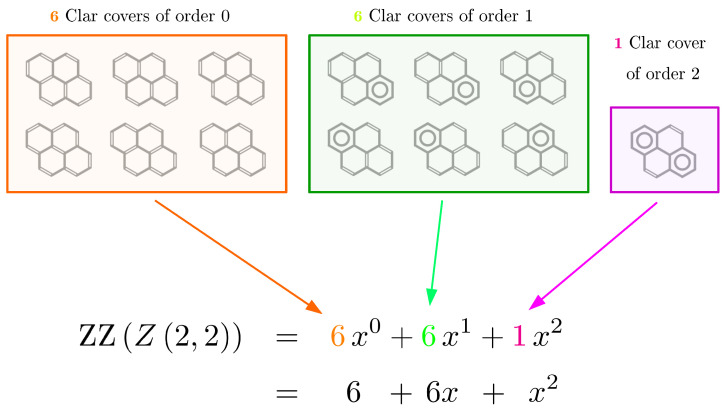
All the 13 conceivable Clar covers of the the multiple zigzag chain $Z\left(2,2\right)$ (i.e., pyrene) can be divided into three classes: (i) Clar covers of order 0 (depicted in the orange frame and usually referred to as the Kekulé structures of $Z\left(2,2\right)$) involve no aromatic rings, (ii) Clar covers of order 1 (depicted in the green frame) involve one aromatic ring, and (iii) Clar covers of order 2 (depicted in the magenta frame) involve two aromatic rings. The associated ZZ polynomial of $Z\left(2,2\right)$, denoted as $\mathrm{ZZ}\left(Z\left(m,n\right),x\right)$, enumerates the number of Clar covers in each class, with the coefficient of each monomial equal to the number of Clar covers in a given class and the power of the monomial equal to the order of a given class.

**Figure 2 molecules-26-02524-f002:**
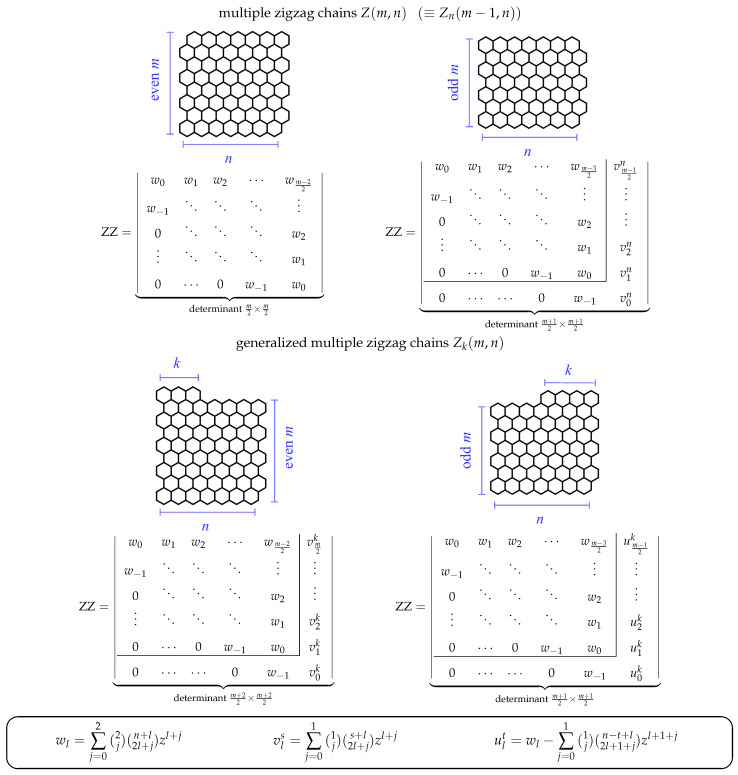
Zhang–Zhang (ZZ) polynomials of multiple zigzag chains $Z\left(m,n\right)$ and generalized multiple zigzag chains $Z_{k}\left(m,n\right)$ can be expressed as determinants of Toeplitz or almost-Toeplitz matrices. The matrix elements are given in the oval frame or by Equations (Equation 28)–(Equation 30).

**Figure 3 molecules-26-02524-f003:**
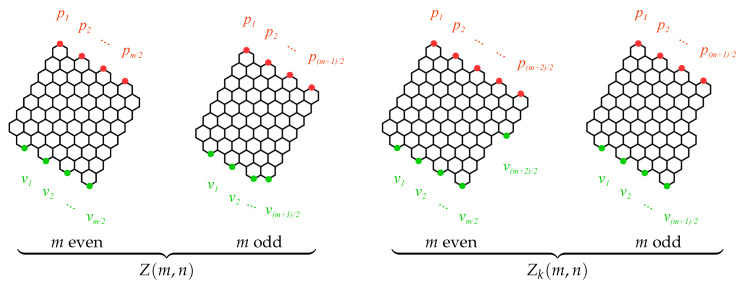
The preferred orientation of the benzenoids $Z\left(m,n\right)$ and $Z_{k}\left(m,n\right)$ allowing to define the optimal sets of peaks $p_{i}$ and valleys $v_{j}$.

**Figure 4 molecules-26-02524-f004:**
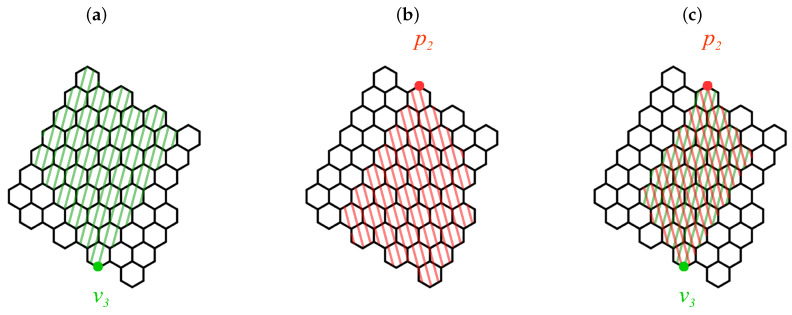
Schematic representations of various concepts related to John–Sachs theory: (**a**) funnel region of the valley $v_{3}$, (**b**) wetting region of the peak $p_{2}$, and (**c**) the $p_{2}$ → $v_{3}$ path region obtained as the intersection of the wetting region of $p_{2}$ and the funnel region of $v_{3}$.

**Figure 5 molecules-26-02524-f005:**
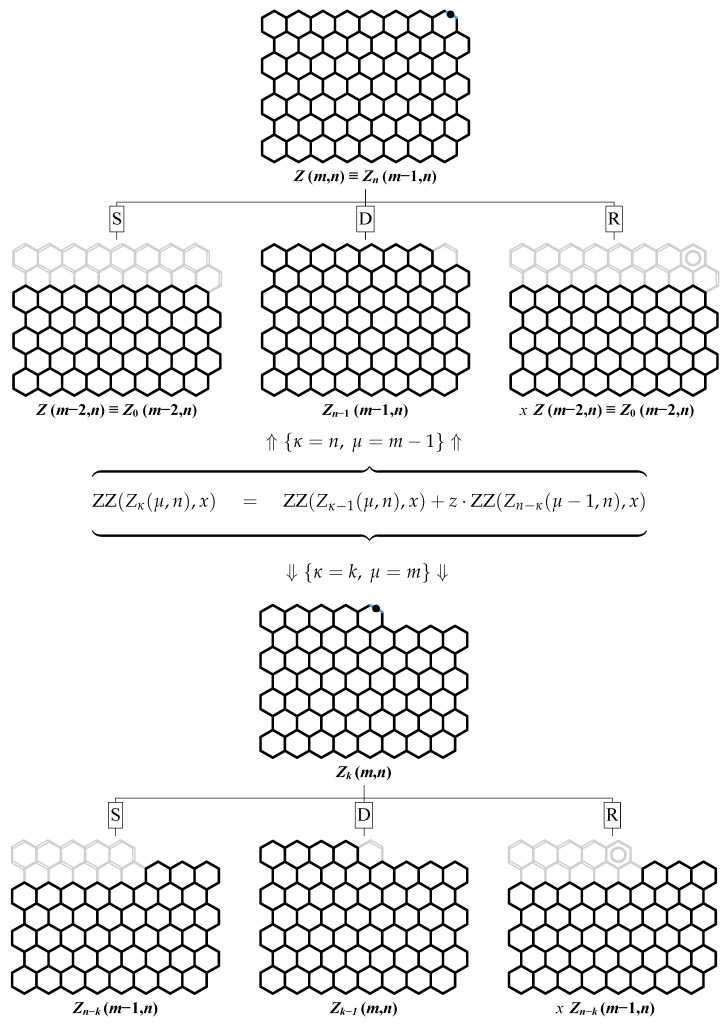
Recursive decomposition of the benzenoids $Z\left(m,n\right)$ (**upper panel**) and $Z_{k}\left(m,n\right)$ (**lower panel**) with respect to the edge depicted in blue and marked with a black dot produces in each case three benzenoids isostructural to $Z\left(m,n\right)$ or $Z_{k}\left(m,n\right)$. Both decompositions can be represented by the same recurrence relation with z=1+x, κ=n, and μ=m−1 for the upper decomposition and κ=k and μ=m for the upper decomposition.

**Figure 6 molecules-26-02524-f006:**
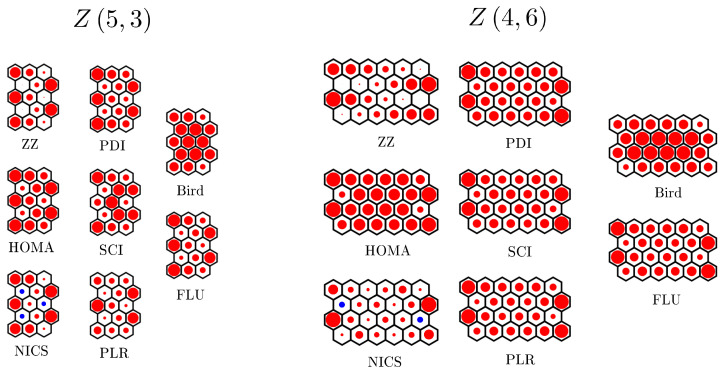
ZZ aromaticity indices computed for all the hexagons of multiple zigzag chains $Z\left(5,3\right)$ (**left**) and $Z\left(4,6\right)$ (**right**) are compared to analogous quantities obtained from rigorous quantum chemical calculations. We use the following quantum chemical aromaticity indicators: NICS [81], HOMA [82,83], Bird [84], PDI [85], SCI [86], PLR [87,88,89,90], and FLU [91]; for more details, see in [92]. The size of the red circle corresponds to the aromaticity of each hexagon, with the largest circle corresponding to the maximal aromaticity obtained with each method. The blue circles in NICS aromaticity patterns correspond to anti-aromatic hexagons with positive chemical shift.

**Figure 7 molecules-26-02524-f007:**
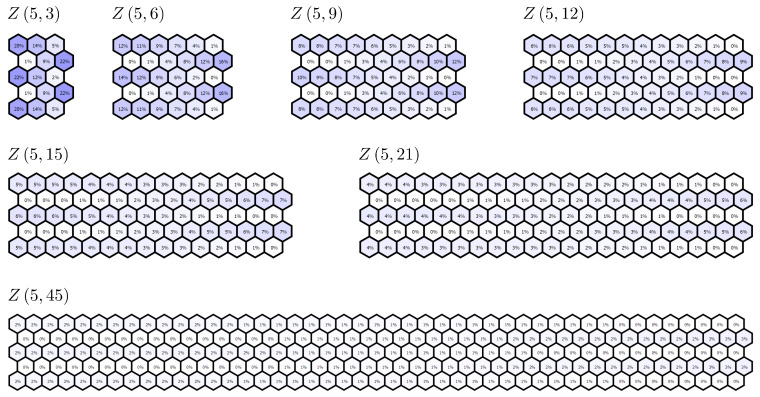
ZZ aromaticity indices computed for all the hexagons of multiple zigzag chains $Z\left(5,n\right)$ for n=3,6,9,12,15,21, and 45. The positions of the most aromatic and most antiaromatic hexagons do not change with the length of $Z\left(5,n\right)$, but the longer flakes loose almost all the aromaticity in comparison with an infinite graphene sheet.

**Figure 8 molecules-26-02524-f008:**
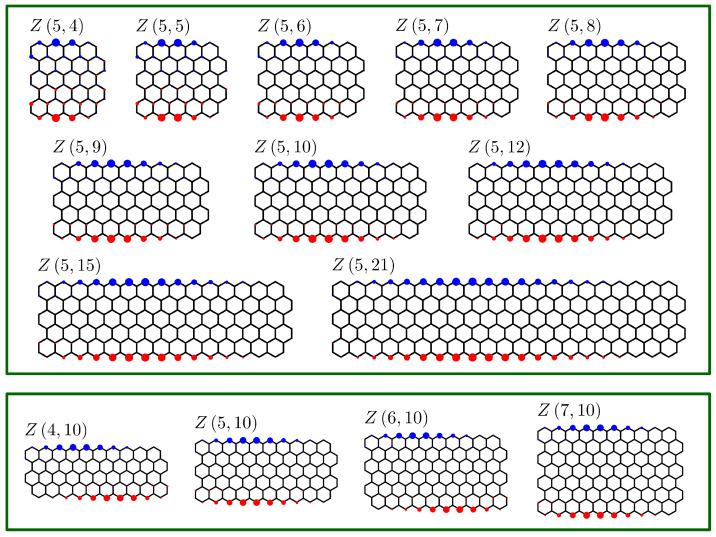
ZZ spin populations on carbon atoms in biradical structures of $Z\left(5,n\right)$ with n=4,5,6,7,8,9,10,12,15 and 21 (**upper panel**) and $Z\left(m,10\right)$ with m=4,5,6 and 7 (**lower panel**). The computed ZZ spin population pattern is composed of two spin waves of opposite signs (depicted in blue and red) located at the zigzag edges of each multiple zigzag chain. The value of the spin population is visualized as a circle centered on each carbon atom. The predicted patterns very closely resemble the antiferromagnetic states of graphene nanoribbons discovered previously in quantum mechanical calculations; for more details, see text.

## Data Availability

The data presented in this study are not available on request from the corresponding author.

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
