# Peer review of "Zhang–Zhang Polynomials of Multiple Zigzag Chains Revisited: A Connection with the John–Sachs Theorem"

_molecules, 2021, doi:10.3390/molecules26092524_

Round 1
Reviewer 1 Report
The paper by Witek reports on Zhang-Zhang polynomials of multiple zigzag chains, which are obtained by exploiting the John-Sachs theorem. The paper is well-written and clear. However, the paper is not recommended for publication in its present form because the chemical insight does not emerge. In particular, the paper is written for a mathematical readerships and other journals are more suited for this purpose. In fact, in my opinion, this work, even if very interesting, falls outside the scope of Molecules and is more suited for a mathematical chemistry journal. To give an example, in the conclusions the author states:
"We hope that the presented here results will stimulate the graph-theoretical community to discover further examples of generalized John-Sachs matrices PZZ also for other benzenoids, which will pave the road to conception and formulation of the generalization of John-Sachs theorem"
I do believe that the graph-theoretical community is not the common readership of Molecules. In order to make it publishable in Molecules, the work should be entirely rewritten by mainly focusing on the chemistry and not on the mathematics.
Author Response
I would like to thank the Referee for a fair and clear review.
I fully understand the Referee's concern about somewhat excessive mathematical content of the presented manuscript from a viewpoint of a chemist. However, I would like to bring to the attention of the Referee an important point: The presented manuscript concerns issues dwelling in the intersection of mathematics and chemistry. Having an almost 10 years experience of publishing in this field, I have a recollection of receiving reviews from chemical journals complaining about "excessive mathematical character" of the presented manuscript, and similar reviews from mathematical journals complaining about "excessive chemical character" of the presented results. In some cases the publication of similar manuscripts was denied even without a review process because of such arguments (including the Journal of Mathematical Chemistry, ironically seeming the most appropriate venue for hosting such work...). But then there is an important question: Where to publish such interdisciplinary results? Which journal is appropriate? There seem to be not a good answer for this question and therefore our papers on ZZ polynomials are published both in chemical and mathematical journals including MATCH Communications in Mathematical and in Computer Chemistry, Discrete Applied Mathematics, Symmetry, Journal of Mathematical Chemistry, and others. In this light, an attempt to publish mathematical theory of graphene nanoribbons in Molecules seems appropriate to me: these structures are indeed highly intriguing from both chemical and mathematical point of view.
In order to make our "chemical" exposition stronger and justify publishing our results in Molecules, we have added to the manuscript a whole new Section titled "Chemical applications" that spans 4 out of 28 pages of the revised manuscript. In this section, we apply the developed mathematical machinery of ZZ polynomials of multiple zigzag chains to predicting aromaticity distribution in finite and infinite nanoribbons and in predicting spin densities in biradical states of multiple zigzag chains. The results are benchmarked against accurate quantum chemical calculations, showing good agreement and similar trends. In addition, our methodology can be applied to much larger structures than those usually treated by quantum chemical methods, including also the transition from molecular to crystal regime, almost impossible to treat formally from the quantum chemical stand point. I believe that the presented chemical applications would be of interest for chemists, and at the same time that they provide a useful and beautiful application of the obtained mathematical results to purely chemical analysis.
I hope that the presented changes would be to the satisfaction of the Referee.
Reviewer 2 Report
The present study is a specific example for the scope and importance of the theoretical chemoinformatics and combinatorial chemistry. It is correctly postulated and performed and the final goal to extend the mathematical analysis to generalized multiple ZZ chains is achieved. Having in mind the specificity of the study performed I recommend acceptance in its present form without revisions.
Author Response
I thank the referee for the favorable review.
Reviewer 3 Report
I recommend that this manuscript should be majorly revised.
In this manuscript, the author pedagogically investigated the Zhang-Zhang (ZZ) polynomials based on a hypothetical extension of the John-Sachs theorem. As a result, he derived ZZ polynomials of zigzag chains using Toeplitzz matrices. Though this may have significant mathematical expressions, it needs more explanations to become publishable as follows:
- This journal (Molecules) targets on the chemistry of molecules. The author, therefore, should detail the significance of the Zhang-Zhang polynomials in the field of chemistry. I guess that it is closely related to the electronic properties of graphite sheets. Several possible applications should also be suggested.\
- Several main technical terms of this manuscript are not sufficiently explained: e.g., the Clar covers, the Johh-Sachs theorem, and zigzag chains. These should be detailed with citations to the references.
- The possibility for the fast and robust computation that the author mention in the manuscript is too obscure to be accepted. It should be specifically explained.
- Though the ZZ polynomials are given in the hypergeometric functions that are too complicated to be intuitively recognized, they could be expressed by standard functions.
Author Response
I thank the Referee for a detailed review. The answers to Referee's comments and the appropriate actions taken to address these comments are discussed below.
- A completely new section has been added to the manuscript to highlight possible chemical applications of the developed mathematical machinery. This new section is now "Section 5. Chemical applications". It consists of 4 pages of text including three figures. We discuss two chemical applications of the derived formulas: (1) determination of aromaticity pattern in finite and infinite multiple zigzag chains and (2) determination of spin densities in finite multiple zigzag chains in states of biradical character. Both applications are compared with accurate quantum chemical calculations. We have selected two multiple zigzag chains, Z(5,3) and Z(4,6), as benchmark structures. The ZZ aromaticity patterns of these two structures agree quite well with quantum chemical predictions. The ZZ aromaticity predictor is applied subsequently to much larger graphene nanoribbons, for which quantum chemical calculations would be either impossible or extremely challenging. In particular, we are able to formally demonstrate that infinite graphene nanoribbons are not aromatic. The second analysis concerns the distributions of two unpaired spins in biradical states of graphene nanoribbons. Our calculations predict that the unpaired spins form two spin waves along each zigzag edge of the graphene nanoribbon. This finding is in close agreement with previous solid state reports of spontaneous magnetism in these materials. It is remarkable that both these properties can be discovered completely without any quantum mechanical calculations, but simply on the base of topological considerations. We hope that these two applications satisfy the Referee and properly address the Referee's comment.
- It is indeed true that the definition of some of the terms was not particularly clear or it comes a bit late in the manuscript. To improve this part of exposition, we have added a new figure in the Introduction section (Fig.1 on page 2), which explains graphically the construction of Clar covers of pyrene, one of the simplest multiple zigzag chains. This figure, together with the text in its caption defines now most of the missing terms highlighted by the Referee. In particular, this figure defines now implicitly Clar covers, the order of Clar covers, the ZZ polynomial, and the relationship between the ZZ polynomial and the set of Clar covers. Concerning the other parts of Referee's remarks, we would like to point out that the sentence "[...] of multiple zigzag chains Z(m,n) and generalized multiple zigzag chains Z_k(m,n). (For a graphical definition of both families of benzenoids, see Fig.3.)" on page 2 of the original manuscript and Fig.3 of the original manuscript (now Fig.4) define the multiple zigzag chains sufficiently well. Similarly, the whole second paragraph of Section 2 "Preliminaries" discusses the John-Sachs theorem, culminating with Eq.(8) giving the mathematical content of this theorem. The main theoretical concepts needed to understand this theorem (i.e., peak, valleys, wetting regions, funnel regions, and path regions) are defined graphically in Figs. 1 and 2 (now Figs. 2 and 3). It seems to us that these concepts are sufficiently clearly defined, but we are open to discuss further possible improvements of the exposition that may make the reading easier for a chemist.
- This comment is somewhat murky and I am not sure I understand entirely its meaning. I believe that the Referee is concerned with the sentence "The new formulas possess both a transparent algebraic form and deep internal structure, which reveal a direct connection to the structural parameters m, n, and k, and offer a possibility for their fast and robust computation." in the Introduction section of the manuscript. It this is indeed so, I agree with the Referee that the word "their" does not fulfill its function here, as it is supposed to mean "ZZ polynomials" and not "the structural parameters m, n, and k". To avoid this grammatical confusion, I have changed the text in this sentence to "The new formulas possess both a transparent algebraic form and deep
internal structure, which reveal a direct connection to the structural
parameters m, n, and k, and offer a possibility for fast and robust determination of the ZZ polynomials in a form of a determinant
of (n+2)-diagonal Toeplitz (or almost Toeplitz) matrices [...]". In practice the algebraic evaluation of the determinant is performed with MAPLE or Mathematica. - I understand the requirement of the Referee to remove the hypergeometric function from the text and replace them by their polynomial representations. Hypergeometric functions are not an element of standard mathematical education of a chemist, or even a physicist nowadays, and their meaning might be somewhat obscure to these two communities. However, a tremendous amount of combinatorial enumeration problems have the most natural solution in hypergeometric functions, and their presence in this manuscript is completely justified in this context. Moreover, the identification of the polynomials in Eq.(13) as 2F1[-2,-n+l;1+2l;z] is almost impossible without the hypergeometric approach. This idea was highlighted in the original manuscript on page 8 right after Eq.(18): "we wnat to stress that the hypergeometric form in Eq.(17) has been essential in the process of the identification of w_l, furnishing a convenient unified framework for the search process." Therefore, I find it very hard to remove the hypergeomtric notation entirely from the manuscript. However, the Referee is right that there is a number of places in the manuscript, where having the choice of using a hypergeometric notation and its elementary function counterpart, the hypergeometric function was selected. This choice can be easily reversed and to satisfy the Referee's requirement, the following places have been modified:
- In text before Eq.(22) "2F1[-1,-k;1,1+x]" has been replaced by its elementary representation "1+k(1+x)".
- In text after Eq.(24) "2F1[-1,k;1;z]" has been replaced by "1+kz".
- In the remaining places, the hypergeometric formulation is always accompanied by the elementary function formulation, with both formulations being simply a translation of each other. This concerns the equations in Eqs.(26) and (27), the equations in Eqs.(28), (29), and (30) and the non-numbered equations in Fig.3 (now Fig.4), the equations in Eqs.(22) and (23), and the equations in Eqs.(17) and (18).
Round 2
Reviewer 3 Report
I think the manuscript is appropriately revised.
So, I recommend it is publishable.